

# Hydrothermal synthesis of carbon and sulfur mono-doped sodium tantalates

Sunil Karna[1], Christopher Saunders[1], Roma Karna[1], Deepa Guragain[2], Sanjay Mishra[2] and Priya Karna[3]

[1] Department of Natural Science, Union College, Barbourville, KY, United States of America
[2] Department of Physics, University of Memphis, Memphis, TN, United States of America
[3] Department of Chemistry, University of Kentucky, Lexington, KY, United States of America

## ABSTRACT

A set of experiments was conducted to synthesize doped and undoped sodium tantalates with carbon and sulfur in energy efficient single-step hydrothermal process. Undoped sodium tantalate nanocubes were synthesized at 140 °C and doped one at 180 °C for 12 h in rich alkaline atmosphere. The sizes of undoped, carbon-doped, and sulfur-doped sodium tantalate nanocubes were 38 nm, 45 nm, and 40 nm, respectively. The morphological, elemental, compositional, structural, thermal, and photophysical properties of as-synthesized doped and undoped sodium tantalate ($NaTaO_3$) were characterized using scanning electron microscope (SEM), energy dispersive x-ray spectroscope (EDS), Raman spectroscopy, X-ray powder diffraction (XRD), thermal gravimetric analysis (TGA), Fourier transform infrared spectrophotometer (FTIR), and UV-vis spectrophotometer. The sulfur doped $NaTaO_3$ shows a higher photocatalytic activity in degradation of methylene blue than carbon doped and the undoped $NaTaO_3$. The band gaps of undoped $NaTaO_3$, carbon doped c-$NaTaO_3$, and sulfur doped s-$NaTaO_3$ were calculated to be 3.94 eV, 3.8 eV, and 3.52 eV, respectively using Tauc plot.

## INTRODUCTION

Sodium tantalates are perovskite compounds of sodium bonded with tantalum and oxygen atoms with definite proportion. A material that obeys the crystallographic structure of calcium titanate ($CaTiO_3$) is usually known as perovskite material. The perovskite structure (usually $ABC_3$ type) simply consists of a large cation A with a 12-fold coordination at the center of a cubic lattice. The corners of the cube are relatively smaller cation B with 6-fold coordination, and the midpoint of each edge are occupied by smaller anions C (halides or oxides). Alternately, large crystal system has cations A at corners, cation B at the center of the cube, and anions C or ($O^{2-}$) in the middle of each face as shown in Fig. 1 (*Ward, 2005*). In Fig. 1, Na stands for A, Ta stands for B, and O stands for C. A unit cell of $NaTaO_3$ shown in Fig. 1A has Ta at octahedron center and the Fig. 1B has $Na^+$ shared with 12 $O^-$ ions of 8 octahedra. Figure 1C is a probable molecular structure of sodium tantalate (*Database, 2008*).

Corresponding author
Sunil Karna, skarna@unionky.edu

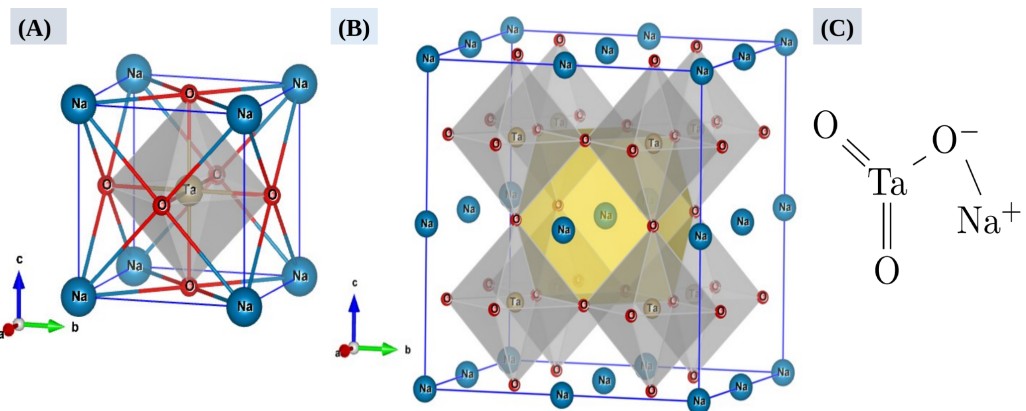

**Figure 1** **Perovskite crystal structure of NaTaO₃ (A), a 3D framework of corner sharing [TaO₆] octahedra with Na⁺ ions in the twelve-fold cavities in between the polyhedral (B), and its molecular structure (C).** Crystal structure of NaTaO₃ was plotted using VESTA-software (*Momma & Izumi, 2011*).

Perovskites with transition metal ions at site B exhibit interesting physical properties because of the distortion in crystals made by the dipole moment of central cation B (*Ward, 2005*; *Johnsson & Lemmens, 2007*). The distortion in an ideal cubic form of perovskite resulted in orthorhombic, rhombohedral, hexagonal, and tetragonal forms. The structural evolution and change in electronic structure of a compound are responsible for their functional properties. These functionalities can be utilized in catalysis, fuel cells, and electrochemical sensing (*Ward, 2005*; *Johnsson & Lemmens, 2007*; *Gregory et al., 2002*; *Okoye, 2005*; *Atta, Galal & El-Ads, 2016*).

Metal oxide based nano perovskite exhibits a high activity for the photocatalytic decomposition of water and photodegradation of organic pollutants under ultraviolet irradiation, which could help minimize the environmental pollution and find alternative energy resources (*Smith, 2013*). The photocatalyst such as NaTaO₃ utilizes only few fractions of visible light spectrum for useful work because of its wide band gap of about 4 eV. Fortunately, it has been observed that the features of NaTaO₃ can be tuned by doping of cations or anions into the stoichiometric phase of its base structure to change its physical and chemical behavior (*Li & Zang, 2009*; *Kang & Park, 2010*; *Li et al., 2015*; *Kanhere & Chen, 2014*; *Lan et al., 2015*). Doping of metal cations or anions on NaTaO₃ reduces its band gap and enhances the visible light response. The process of cation doping on NaTaO₃ are, however, a bit complicated and would limit the economical utilization of the photocatalyst (*Kang & Park, 2010*; *Li et al., 2015*). Anion doping into the oxygen site of metal oxides can be useful and easily controlled during synthesis process. Thus, photocatalytic activities of NaTaO₃ can also be enhanced by doping of non-metal anions such as nitrogen, sulfur, or carbon (*Kanhere & Chen, 2014*; *Lan et al., 2015*; *Liu et al., 2019*; *Shi & Guo, 2012*; *Kudo et al., 2007*; *Fu et al., 2008*). The doping of anions in metal oxides forms an acceptor level because of its electronegativity which increases the valance band-edge potential and thus reduces the band gap. The band gap of the metal oxides reduces significantly if the dopant has P-orbital energy higher than the $O_{2P}$ orbitals

(*Li et al., 2015*). The sulfur has low electronegativity than oxygen, but a higher electron affinity due to its larger size. Sulfur is also a divalent atom that can replace oxygen atom at substitutional site. Such property of sulfur makes it an efficient dopant of $NaTaO_3$. Carbon is also less electronegative than oxygen, but it can contribute more holes in valance band. The C dopants promote separation of photogenerated electrons-holes and thus reduce the rate of recombination (*Lavand & Malghe, 2015*). The S and C are thus considered to be promising dopants for visible light induced photocatalysis. It has also been reported that the dopant like N (nitrogen) leaves excessive numbers of holes in valance band which forms the recombination centers for electron–hole trapping, thus reduces the lifespan of photogenerated charge carriers (*Wang et al., 2013*).

Appreciable amount of work has been done in mono-doping or co-doping of $NaTaO_3$ with dopants like La, Co, Sm, Bi, N, P, and F but a very little work has been devoted to the C or S doping. Hence, through this experiment we have synthesized and doped sodium tantalates ($NaTaO_3$) with C and S to study their comparative photophysical properties. An environmental-friendly low temperature chemical process has been used to synthesize all the samples in this study. In order to optimize growth parameters of the nanostructures we have characterized them for morphological, compositional, structural, thermal, and optical properties using SEM, EDS, Raman, FTIR, XRD, TGA, and UV-Vis Spectroscopy.

## MATERIALS & METHODS

### Materials synthesis

There are various factors that affect the structure of perovskite crystals such as increase of temperature, particle size, tiltation of octahedral, synthesis process, concentration of reaction mixture, and reaction time (*Ahmad, Farooq & Phul, 2018*). Various methods have already been developed to synthesize perovskite materials such as ball milling, thermal evaporation, sol–gel process, and hydrothermal process (*Ward, 2005*). Among all these processes, hydrothermal process is one of the most suitable chemical processes in terms of energy consumption and environmental friendliness for the varieties of perovskite compounds. Size and chemical compositions of oxides type perovskite in hydrothermal process (HTs) can be controlled by adjusting the concentration of precursors, reaction time, and temperature. This process is based on a dissolution/precipitation mechanism. We have used a low temperature HTs to synthesize and dope sodium tantalates ($NaTaO_3$) by optimizing the concentration of the reactant precursors.

The $NaTaO_3$ perovskite were synthesized by reacting tantalum penta-oxide precursor $Ta_2O_5$ in high alkaline environment NaOH under hydrothermal conditions at low temperature. The reaction mechanism is given as

$$2NaOH + Ta_2O_5 \xrightarrow{t^oC} 2NaTaO_3 + H_2O. \tag{1}$$

For sodium tantalate ($NaTaO_3$), 0.442 g of $Ta_2O_5$ powder was dissolved in 0.75 M of NaOH for 5 h with magnetic stirring. The 50 mL solution was then kept in a 100 mL capacity teflon lined autoclave and heated for 12 h at 140 °C (*Li & Zang, 2009*). The autoclaves were naturally cooled down to the room temperature before milky-white product was collected, centrifuged washed with water and ethanol many times, and dried at 80 °C overnight.

For C-doped sodium tantalate (c-NaTaO$_3$), 2g of glucose (C$_6$H$_{12}$O$_6$) was dispersed in a mixture of 30 ml deionized water and 20 ml polyethelene glycol (400) for 15 min with ultrasonication. Then, 1.5 g (0.75M) of NaOH and 0.442 g of Ta$_2$O$_5$ powder were added in the mixture and stirred on a magnetic stirrer for 5 h. The solution was then kept in teflon lined autoclave and heated for 12 h at 180 °C (*Kang & Park, 2010*; *Wu et al., 2014*). The same cleaning procedure as above was used to collect light-brownish product.

For S-doped sodium tantalate (s-NaTaO$_3$), 0.442 g of Ta$_2$O$_5$ powder and 0.200 g of sodium thiosulfate (Na$_2$S$_2$O$_3$:5H$_2$O) were dissolved in 1.5 g of NaOH and 50ml deionized water by magnetic stirring for 5 h and 5 min of ultrasonication. The solution was then kept in teflon lined autoclave and heated for 12 h at 180 °C (*Li et al., 2015*). The same cleaning procedure as previous cases was used to collect white product. The reagent grade tantalum pentaoxide powder bought from Sigma Aldric was used in this experiment.

## Photocatalytic test

The photocatalytic activities of doped and undoped NaTaO$_3$ were evaluated by degrading aqueous solution of methylene blue (MB) in visible light irradiation. The photo reactor chamber was locally improvised where the solution was kept cool using water circulating jacketed beaker of 100 ml and was being magnetically stirred during the process. The stock solution of methylene blue of concentration 20 mg/L was prepared by dissolving 20 mg of methylene blue powder in 1 l of deionized water and stored in dark room for future use.

As synthesized photocatalyst powder of 0.05 g was dispersed in 50 ml of 20 mg/L concentrated methylene blue (MB) aqueous solution. The suspension was then ultrasonicated for 10 min and stored in dark for 30 min to allow adsorption–desorption equilibrium between photocatalyst and the MB. The suspension was being stirred magnetically and kept cool during this process. The white light LED of 50 W was taken as a source of visible light and was kept at about 20 cm above the solution. The suspensions and the aqueous MB solution were kept in dark for 30 min. The UV/Vis spectra of these solutions were taken before and after visible light irradiation. At every 50 min of irradiation about 2 ml of solution was withdrawn from the reactor cell (beaker), centrifuged for 10 min at 3000 rpm and the supernatant of the solution was taken for UV/Vis spectrum. The percentage photo degradation, $\eta$ of MB was taken as

$$\eta = \frac{A}{A_o} \times 100\% \tag{2}$$

where A$_o$ is the absorbance before irradiation, and A is the absorbance obtained after every 50 min of irradiation of sample in visible light. The intensity of light on the exposed surface of MB was calculated to be about 10 mW/cm$^2$.

## RESULTS AND DISCUSSION

The table-top scanning electron microscope (SEM) was used to study the morphology of the samples. The cubic morphology of undoped and doped nanostructures is shown in Figs. 2A–2C. The morphology of the samples seem similar may be because of the range of doping is very little as can be depicted by XRD of the samples. However, the morphology

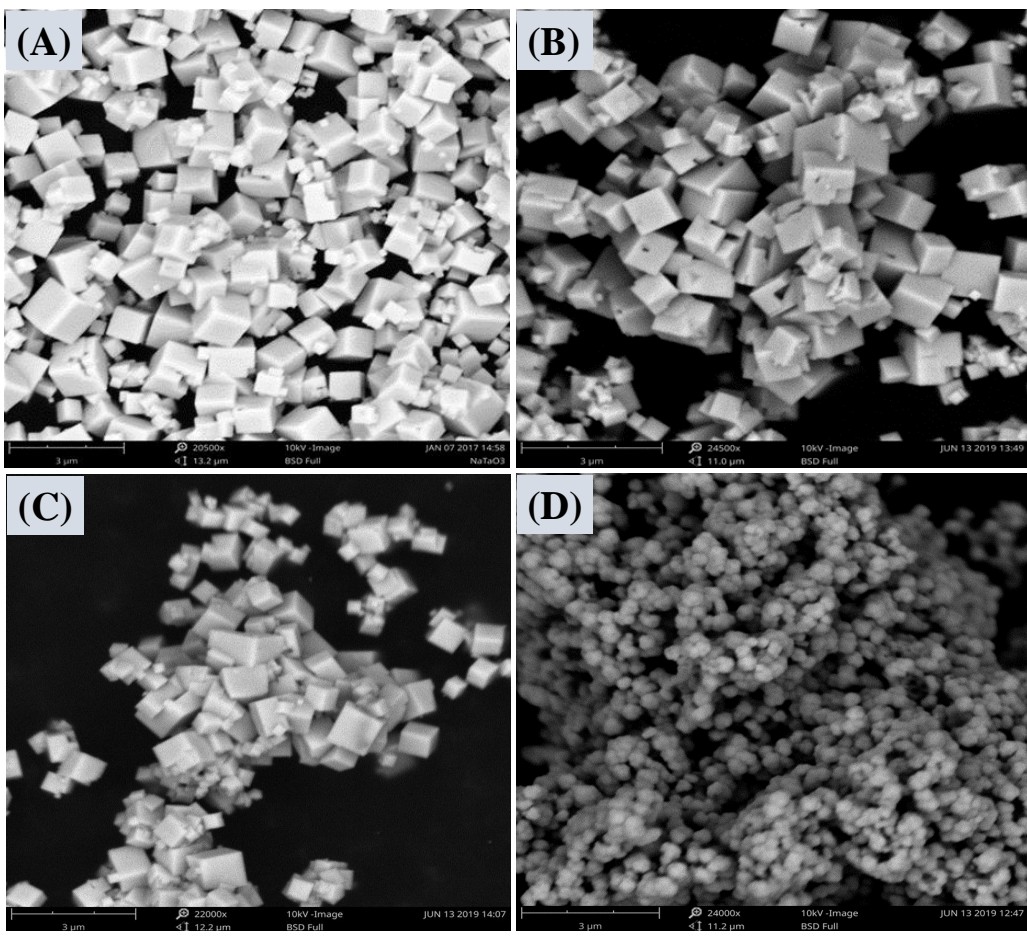

**Figure 2** **SEM images of the pure NaTaO₃ (A), c-NaTaO3 (B), s-NaTaO ₃ (C), and c-NaTaO₃ without glucose as a reaction component (D).**

of carbon doped sodium tantalates without using glucose in a reaction mixture along with ethylene glycol during synthesis is shown in Fig. 2D.

The compositions of catalyst are shown in Fig. 3 by the EDS images. The plot indicates presence of dopants such as Sulfur and Carbon effectively incorporated into the sodium tantalate matrix during hydrothermal process.

Raman spectra were taken to verify EDS and XRD study. The bands between 400 to 1,100 $cm^{-1}$ related to the internal vibrational mode of $Ta_2O_6$ in $NaTaO_3$ structure (*Vishnu et al., 2008*; *Hernandez, Flores & Martinez, 2018*). The major bands at 450, 500, 620, and 720 $cm^{-1}$ in $NaTaO_3$ are reported earlier (*Longjie & Hiroshi, 2015*). One additional band appeared at 860 $cm^{-1}$ which is associated with doping of sulfur and may be related to the phase transition of $Ta_2O_6$ octahedra tiltation as shown in Fig. 4A. In Fig. 4B, the broad mask between 1,100 to 1,800 $cm^{-1}$ engulfing the characteristics D and G bands is related to the amorphous carbon and explains the formation of composite between carbon and

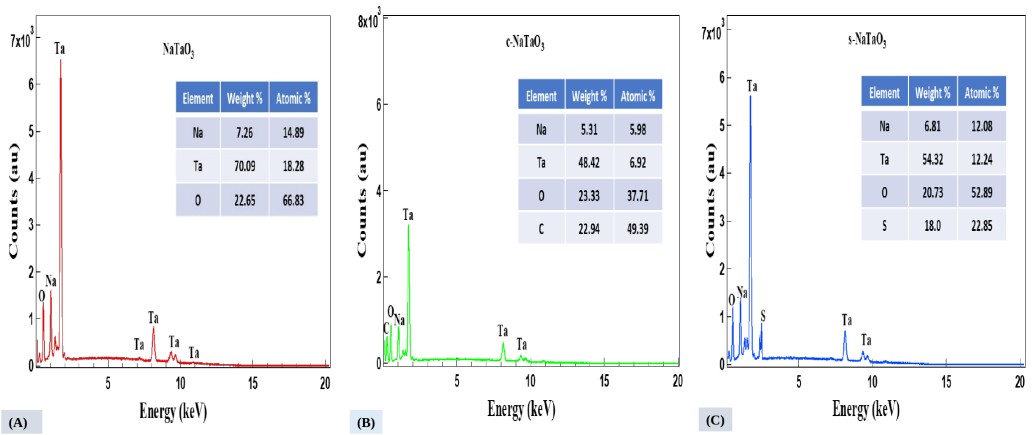

**Figure 3** EDS spectra of NaTaO₃ (A), c-NaTaO₃ (B), and s-NaTaO₃ (C), respectively.

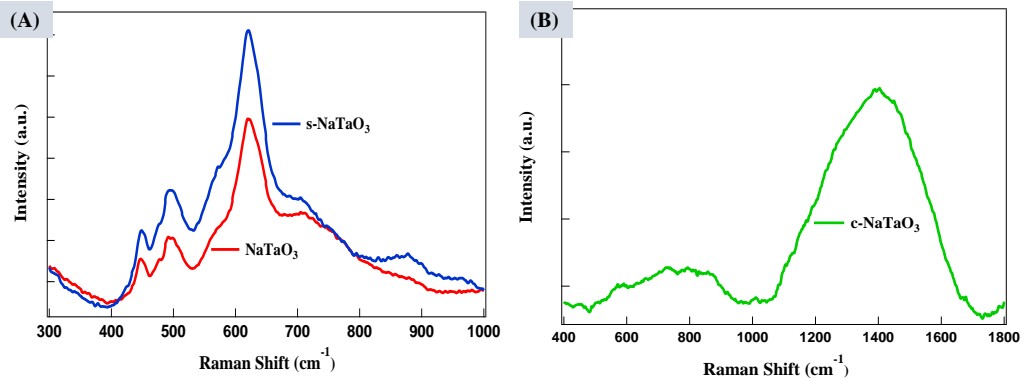

**Figure 4** Raman spectra of (A) NaTaO₃ and s- NaTaO₃ and (B) c- NaTaO₃ samples.

NaTaO₃ (*Hernandez, Flores & Martinez, 2018*). The excessive amount of carbon in host material acts as a recombination center and reduces the photocatalytic activity.

Bruker D8 Advance X-ray Diffractometer was used to collect X-ray diffraction patterns. XRD patterns of sodium tantalate (NaTaO₃) crystals along with their Rietveld refinement profiles are shown in Figs. 5A and 5B. The observed and calculated XRD profiles of NaTaO₃ with the help of crystallography open database (COD) are seen in fair agreement with each other and suggests single phase compounds of cubic perovskite structure with space group Pm-3m (#221). The lattice parameters, space group, and atomistic positions were analyzed using Rex, Maud, and PowderX software from observed XRD data. The crystal data and refinement factors of the sample NT (NaTaO₃) are summarized as Lattice Parameters: $a = b = c = 3.89002$ Å, $\alpha = \beta = \gamma = 90^\circ$, $R_p = 0.16$, $R_{wp} = 0.23$, $R_{exp} = 0.18$, and GoF $= 1.27$. Undoped (NaTaO₃), carbon doped (c-NaTaO₃), and sulfur doped (s-NaTaO₃) sodium tantalates are shown in Fig. 5C with inset showing prominent peaks are slightly shifted towards lower $2\theta$ angle as doped. However, C-doped sample has not shown any

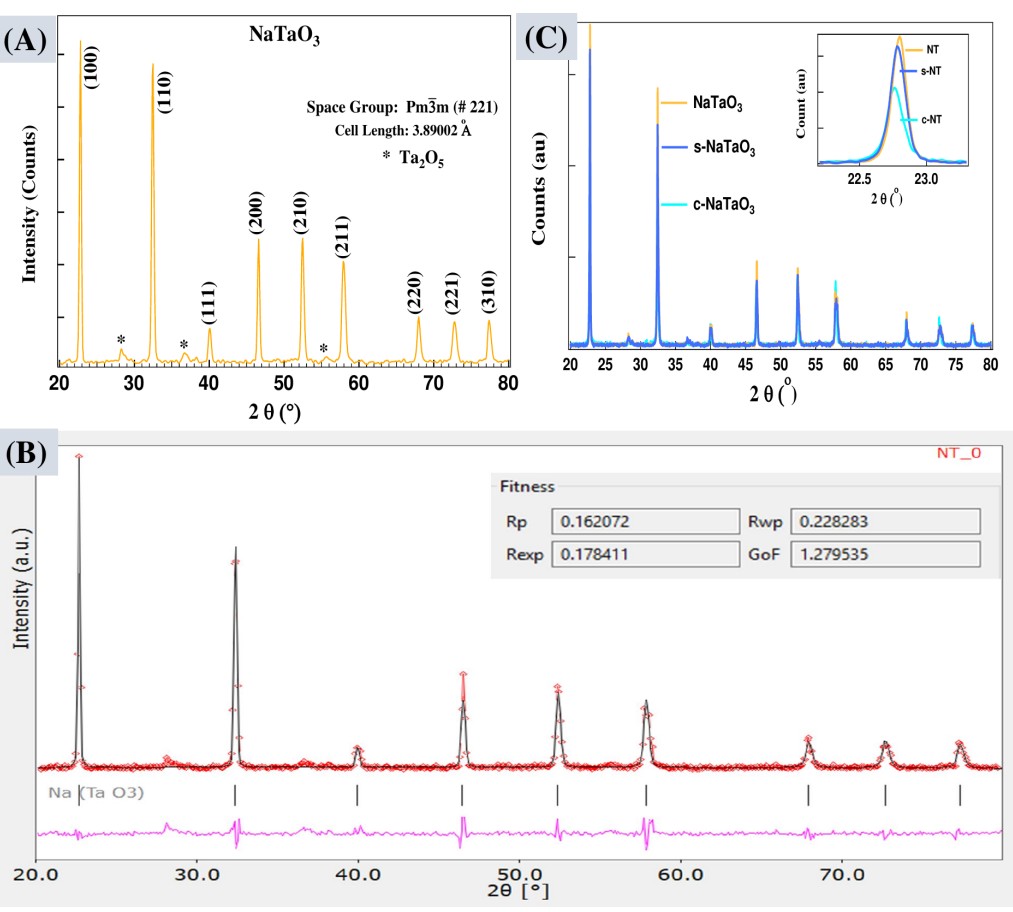

**Figure 5** XRD patterns of NaTaO₃ (A), Rietveld refinement profile of the sample NT using Rex software. Experimental pattern (red), calculated data fit (cyan), and difference curve are shown in pink in the refined data (B), XRD pattern of all the samples.

noticeable shift which could be due to formation of amorphous carbon in the host materials as can be verified by Fig. 4B. The crystal data and refinement factors of the sample c-NT (c-NaTaO₃) are $a = b = c = 3.90262$ Å, $\alpha = \beta = \gamma = 90°$, $R_p = 0.27$, $R_{wp} = 0.39$, $R_{exp} = 0.30$, and GoF = 1.28, and that of s-NT (s-NaTaO₃) are $a = b = c = 3.89830$ Å, $\alpha = \beta = \gamma = 90°$, $R_p = 0.28$, $R_{wp} = 0.47$, $R_{exp} = 0.30$, and GoF = 1.53, respectively.

The average crystallites sizes (L) of NaTaO₃, s-NaTaO₃, and c-NaTaO₃ are 38 nm, 40 nm, and 45 nm, respectively as measured from full width at half maximum (FWHM) of prominent XRD peaks at about 32° of 2θ peak position using Scherrer's formula (*Langford & Wilson, 1978*),

$$L = \frac{K\lambda}{\beta \cos(\theta)},$$ (3)

where $K = 0.89$ for cubical symmetry, FWHM (β) at 2θ in radian unit, and $\lambda = 0.1540598$ nm. XRD data indicates highly crystalline cubic nanoparticles with d value of (100) plane is about 0.3903 nm. A detailed inspection of peaks displays the slight shift of position

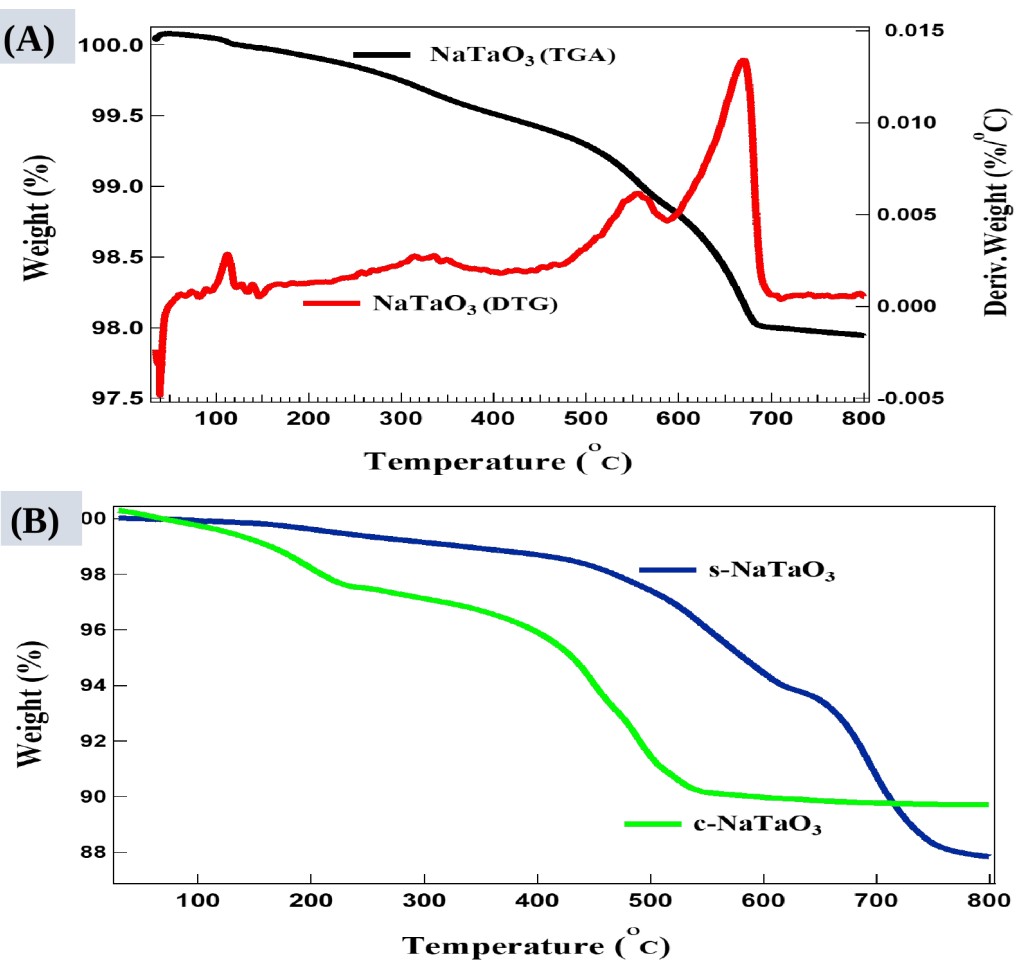

**Figure 6    TGA/DTG curves of NaTaO3 (A), c-NaTaO3 and s-NaTaO3 (B). .**

towards lower angle in doped samples, indicating the tendency of modifying cubic phase of NaTaO$_3$ crystal.

TGA curves were obtained in nitrogen atmosphere at a heating rate of 10 °C/min as shown in Fig. 6. TGA curve determines thermal stability and monitors decomposition behavior of synthesized nanocubes. Each sample produces different sigmoidal shapes of TGA curves. Gradual but slow weight loss up to 150 °C indicates dehydration and evaporation of volatile contents in the sample (such as residual ethanol contents from cleaning). A little incremental weight below 100 °C may be due to buoyancy effect of atmosphere inside the TG system. The samples were hygroscopic in nature due to the synthesis process, and it is expected to have weight loss below 200 °C. The higher weight loss was observed between 300–700 °C associated with residual low molecular weight substance, decomposition reaction, and coordinated water elimination. Since TGA curves are dependent on heating rate hence any experimental parameters that effect the reaction rate will change the shape of curve e.g., type of materials, purge gas, sample mass, volume, and morphology (*Froberg, 2020*). Sodium tantalate, NaTaO$_3$ sample has shown very little

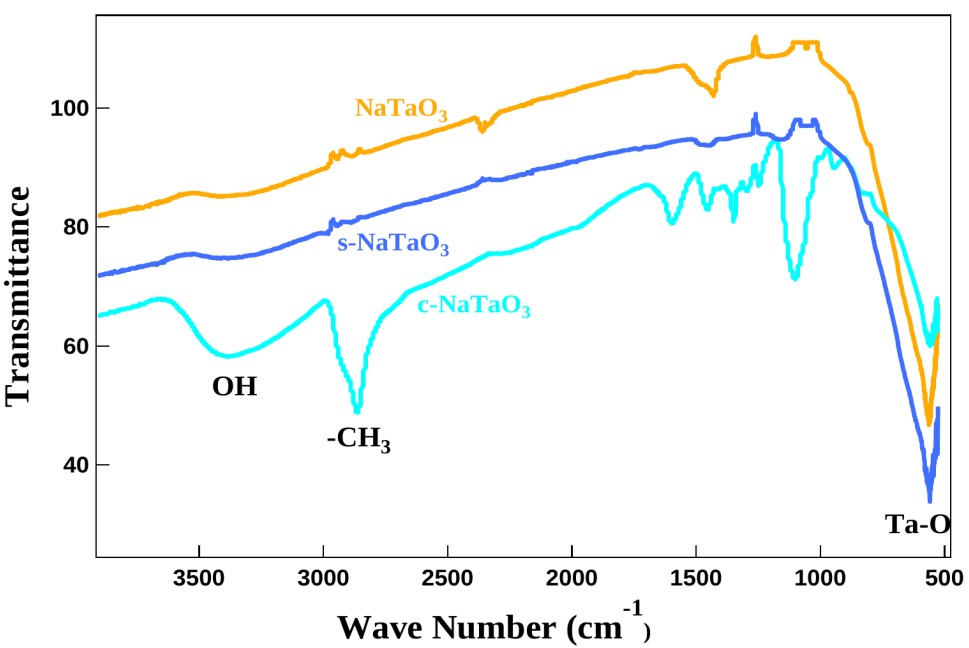

**Figure 7** **FTIR curves of doped and undoped NaTaO3.** The plot is offset by 10 points for clarity.

weight loss altogether of about 2.37%. The first derivative of TGA curve (DTG) reveals several steps of mass loss. The DTG curve shows that 0.87% and 0.52% mass loss happen at 660 °C and 550 °C, respectively. The peak at 110° C corresponds to evaporation of volatiles as shown in Fig. 6A. The TGA of s-NaTaO$_3$ and c-NaTaO$_3$ samples in Fig. 6B has shown the high degradation of impurities from sodium thiosulphate residue and the glucose residue and cause their weight loss of about 12.4% and 10%, respectively. The c-NaTaO$_3$ also shows oxidation process between 250 °C to 460 °C and become thermally stable after 560 °C.

Figure 7 shows FTIR spectra of as synthesized samples. The spectrum of NaTaO$_3$ comprises of a strong band at 560 cm$^{-1}$ which corresponds to Ta - O stretching bond. The spectrum of s-NaTaO$_3$ displays similar spectrum as undoped NaTaO$_3$ without any bands related to SO$_3^{2-}$ or SO$_4^2$- indicating no residue of sulfur during washing. The bands around 1,000 to 1,630 cm$^{-1}$ corresponds to Na - O vibrations. A broad peak around 3,400–3,500 cm$^{-1}$corresponds to H$_2$O bands and strong stretching modes of the O-H band. The intense peak at 1,068 cm$^{-1}$corresponds to CO$_3^{-2}$ vibration band in spectrum of c-NaTaO$_3$ (*Li et al., 2015*).

The visible light photo degradation of MB was analyzed using UV/Vis spectrometer. The peak at 664 nm was chosen to study the degradation of MB with and without catalysts and has been shown in Fig. 8A. Not an appreciable change has been observed in UV/Vis spectra of blank MB solution after keeping 30 min in dark, depicting the reaching of the adsorption desorption equilibrium. The photoactivity of blank MB solution was very low, only 1.7% degradation was observed after 300 min. The degradation of MB and other dyes in visible light photolysis has been reported by previous authors (*Esparza et al., 2020*;

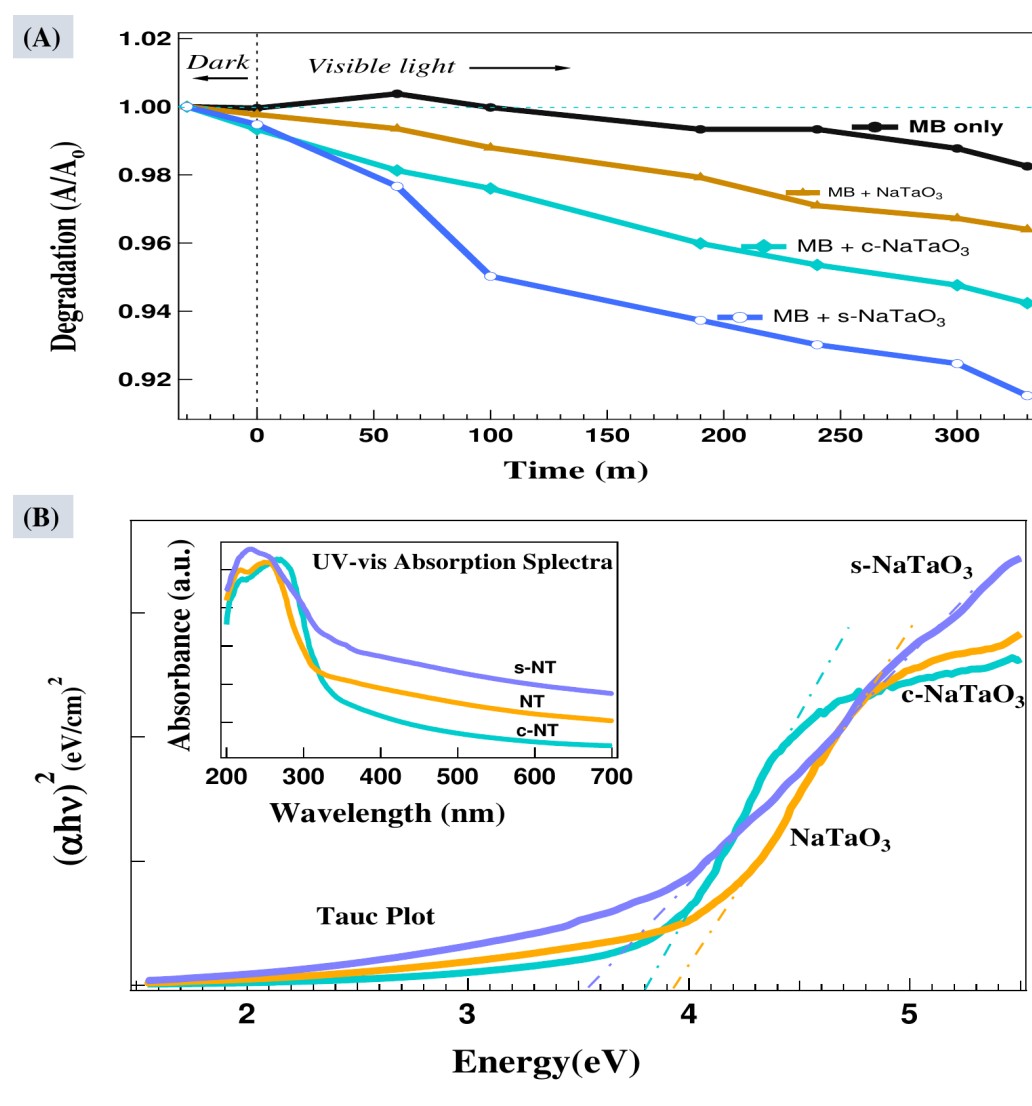

**Figure 8** Photodegradation of methylene blue (MB) in solution containing $NaTaO_3$, c -$NaTaO_3$, and s-$NaTaO_3$ catalysts (A), and Tauc plot with inset UV-vis diffused reflectance spectra of all the samples (C).

*Li et al., 2015*; *Lan et al., 2015*; *Hou, Hu & Zhu, 2018*; *Lavand & Malghe, 2015*; *Khaneghah, Yangjeh & Abedi, 2018*). The MB with catalysts $NaTaO_3$, c-$NaTaO_3$, and s-$NaTaO_3$ has shown 3.4%, 5.1%, and 7.8% degradation, respectively within 300 min of visible light irradiation. Due to large optical band gap of these catalysts and consequently very poor absorption of visible light its degradation efficiency was very poor. Figure 8B shows the UV/Vis absorption spectra of $NaTaO_3$, c-$NaTaO_3$, and s-$NaTaO_3$ catalysts. The Tauc plot of $(\alpha h\upsilon)^2$ ploted against energy from UV/vis diffused reflectance spectra shows the photocatalysts $NaTaO_3$, c-$NaTaO_3$, and s-$NaTaO_3$ have direct band gap energy of 3.94 eV, 3.8 eV, and 3.52 eV, respectively.

## CONCLUSIONS

Sulfur and carbon mono-doped sodium tantalate nanocubes were grown successfully in rich alkaline atmosphere by low temperature hydrothermal process. The sodium tantalate ($NaTaO_3$), carbon doped sodium tantalate (c-$NaTaO_3$), and sulfur doped sodium tantalate (s-$NaTaO_3$) have found perovskite crystal structure of Pm-3m cubic phase with an average size of 38 nm, 45 nm, and 40 nm, and their band gaps were calculated as 3.94 eV, 3.8 eV, and 3.52 eV, respectively. The slight shift in prominent XRD peaks position of S-doped $NaTaO_3$ are found towards lower angle but no noticeable shift has been observed in C-doped $NaTaO_3$. Bandgap of S-doped $NaTaO_3$ is found narrower than C-doped and undoped $NaTaO_3$. S-doped $NaTaO_3$ shows comparatively higher visible light photocatalytic activity than C-doped and undoped $NaTaO_3$. Due to high band gap the photocatalytic activities of these photocatalysts were poor, however, enhancement in adsorption of methylene blue solution has been observed with the use of these photocatalysts.

## ACKNOWLEDGEMENTS

We would like to thank Department of Natural Science and the Physical Plant of Union College for their support in this project.

### Funding

This work was supported by the Faculty Research Committee, and Title III grant, Union College, Barbourville, KY. The funders had no role in study design, data collection and analysis, decision to publish, or preparation of the manuscript.

### Grant Disclosures

The following grant information was disclosed by the authors:
The Faculty Research Committee, and Title III grant, Union College, Barbourville, KY.

### Competing Interests

The authors declare there are no competing interests.

### Author Contributions

- Sunil Karna conceived and designed the experiments, analyzed the data, prepared figures and/or tables, authored or reviewed drafts of the paper, and approved the final draft.
- Christopher Saunders performed the experiments, prepared figures and/or tables, and approved the final draft.
- Roma Karna performed the experiments, prepared figures and/or tables, authored or reviewed drafts of the paper, and approved the final draft.
- Deepa Guragain performed the experiments, prepared figures and/or tables, and approved the final draft.
- Sanjay Mishra analyzed the data, authored or reviewed drafts of the paper, and approved the final draft.

- Priya Karna performed the experiments, authored or reviewed drafts of the paper, and approved the final draft.

### Data Availability

The raw measurements are available in the Supplemental Files.

### Supplemental Information

Supplemental information for this article can be found online at http://dx.doi.org/10.7717/peerj-matsci.10#supplemental-information.

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
