# Peer review of "Hydrothermal synthesis of carbon and sulfur mono-doped sodium tantalates"

_PeerJ Materials Science, doi:10.7717/peerj-matsci.10_

## Round 0.1 · original submission · Major Revisions

1. Equation number is required in the equation provided in Material synthesis section.

Reviewer 1 ·

Basic reporting

No Comments

Experimental design

No Comments

Validity of the findings

No Comments

Additional comments

The manuscript discusses the sulfur and carbon mono-doped sodium tantalate nanocubes in rich alkaline atmosphere by low temperature hydrothermal process. Sodium tantalate, carbon doped sodium tantalite and sulfur doped sodium tantalate were found to have perovskite cubic crystal structure (Pm-3m space group) with an average size of 38 nm, 45 nm and 40 nm, and band gap values of 3.94 eV, 3.8 eV and 3.52 eV respectively. The manuscript is written well and composed of the justified studies, however, it would be good addition if authors may justify the sulphur and carbon doping using XPS studies and changes in Raman studies (if possible) over sodium tantalates. Lack of some relevant references could be seen whose addition may increase the impact of the studies such as Industrial & Engineering Chemistry Research, 57 (1), 18-41, 2018; ACS Omega, 4, 19408-19419, 2019; Scientific Reports, 9, 4488 (1-17), 2019; Desalination and Water Treatment, 162, 303-312, 2019. The manuscript may be considered after addressing the above issues.

Reviewer 2 ·

Basic reporting

NO

Experimental design

Synthesis is not controlled.

Validity of the findings

Poor validity of the findings are present

Additional comments

This work presents the hydrothermal synthesis of photocatalytic C and S doped NaTaO3 for MB degradation. SEM, EDS, XRD, UV-Visible and TGA techniques are utilized as characterization. However, the characterization techniques are insufficient to confirm the doping of C and S in NaTaO3. Lack of evidences has been observed with poor validity of interpreted results. The degradation results ambiguous and lacks with accuracy. I hereby reject the paper in its present form. However, suitable rebuttle of comments, one can acces for its publication.
1. License number of VESTA-software is required
2. EDS analysis required the atomic percent ratios of the synthesized photocatalysts.
3. XPS, Raman analyses or alternative techniques are required to support the doping of C and S successfully as the XRD is the only technique used for the purpose and it seems insufficient.
4. Line 140 stoke solution should be stock solution
5. What is the intensity used for the source light and how is it calibrated?
6. In XRD, C doping should shrink the lattice structure and S should Expand the lattice structure of NaTaO3 but the peaks shiftings are not showing the stated trend and not confirming the doping in lattice structure but may be at the interstices.
7. Thermal gravimetric analysis (TGA) and DTA in Figure 15 shows the degradation of impurities from the glucose residue and sodium thiosulphate residue and may refer as impurity not doping!
8. In figure 17, how can the MB degrade without catalyst? In other words, why balnk run showing degradation?
9. In any case if it shows degradation, will the blank run effects or data was subtracted from the photocatalysts data?
10. PL and IPCE studies should also be present.

---

## Round 0.2 · accepted · Accept

Authors have successfully responded to all reviewers comments. Hence the manuscript is now acceptable for publication.

Reviewer 1 ·

Basic reporting

No Comments

Experimental design

No Comments

Validity of the findings

No Comments

Additional comments

No Comments